# Material Design Methodology for Optimized Wear-Resistant Thermoplastic–Matrix Composites Based on Polyetheretherketone and Polyphenylene Sulfide

**DOI:** 10.3390/ma13030524

**Published:** 2020-01-22

**Authors:** Sergey V. Panin, Boris A. Lyukshin, Svetlana A. Bochkareva, Lyudmila A. Kornienko, Duc Ahn Nguyen, Le Thi My Hiep, Iliya L. Panov, Nataliya Y. Grishaeva

**Affiliations:** 1Lab. of Mechanics of Polymer Composite Materials, Institute of Strength Physics and Materials Science SB RAS, 634055 Tomsk, Russia; lba2008@yandex.ru (B.A.L.); svetlanab7@yandex.ru (S.A.B.); rosmc@ispms.ru (L.A.K.); anohina@mail2000.ru (N.Y.G.); 2Department of Materials Science, Engineering School of Advanced Manufacturing Technologies, National Research Tomsk Polytechnic University, 634030 Tomsk, Russia; nda.ttndvn@gmail.com (D.A.N.); myhiepru@gmail.com (L.T.M.H.); 3Department of Mechanics and Graphics, Tomsk State University of Control Systems and Radioelectronics, 634050 Tomsk, Russia; panov.iliya@mail.ru

**Keywords:** polymer matrix composites, chemical composition, computer simulation, physical experiment, mechanical properties, material design methodology

## Abstract

The main goal of this paper is to design and justify optimized compositions of thermoplastic–matrix wear-resistant composites based on polyetheretherketone (PEEK) and polyphenylene sulfide (PPS). Their mechanical and tribological properties have been specified in the form of bilateral and unilateral limits. For this purpose, a material design methodology has been developed. It has enabled to determine the optimal degrees of filling of the PEEK- and PPS-based composites with carbon microfibers and polytetrafluoroethylene particles. According to the results of tribological tests, the PEEK-based composites have been less damaged on the metal counterpart than the PPS-based samples having the same degree of filling. Most likely, this was due to more uniform permolecular structure and greater elasticity of the matrix. The described methodology is versatile and can be used to design various composites. Its implementation does not impose any limits on the specified properties of the material matrix or the reinforcing inclusions. The initial data on the operational characteristics can be obtained experimentally or numerically. The methodology enables to design the high-strength wear-resistant composites which are able to efficiently operate both in metal–polymer and ceramic–polymer friction units.

## 1. Introduction

The first attempts to design materials with specified properties have been made for improvement of concrete structures through steel reinforcement oriented in the directions of the highest stresses. Later, this approach has been widely used to design composite materials. It enables to change strength, thermophysical, electrical, and other properties of materials.

A significant variety of structures and properties of the commercially available composites have caused development of numerous mathematical models and methodologies for prediction of their characteristics [1,2,3,4]. Theoretical issues and tasks that they solve are broadly divided into two classes:I.Simulation problems: Operational properties of the materials are governed by their chemical composition, volume distribution of phases and their characteristics, type of interface interaction, etc. The listed properties are a set of control parameters determined experimentally. At present, various numerical methods are often used to predict them due to development of computer technologies that enable to solve large systems of equations [5,6,7,8]. These methods are based on variational principles and subsequent averaging of obtained values over a representative volume.II.Design issues: They are inverse simulation problems. When solving them, it is necessary to determine a set of control parameters that provides the material with the specified properties formulated as unilateral (“no more” or “no less”) or bilateral limits (“from and to” or “plus or minus value”).

Examples on material modeling and designing can be found elsewhere [9,10,11,12,13,14]. Optimization methods have been used in most cases when solving such problems [11,12,13,14]. They include multi-objective optimization, which makes it possible to determine the optimal material composition, structure dimensions, technological parameters, etc. [12,13,14].

Integration of the computer simulation and optimization methods has resulted in optimization software development (OptQuest, OPTIMIZ, Ltd.SimRunner2, WITNESS Optimizer, etc.). In most of them, evolution strategies, genetic algorithms, and neural networks have been implemented for decision procedures [12]. These methods are general global optimization algorithms. In doing so, quasi-optimal solutions can be found in minimal time through their use. The Taguchi method has been used to screen variables [13,14]. In particular, effectiveness of integrating the Taguchi and contour curve methods has been shown in [14].

Based on the mentioned papers, we can conclude that a computer-aided design methodology for polymer materials having specified properties is relevant. Their main difference from the conventional optimization problems is necessity to find a prescribed (not the extreme) value (or range of values) of the parameter. For example, the contour curve matching method has been applied there. The use of limited experimental data sets and computer simulation without determination of objective functions make it major advantage [15,16].

## 2. Formulation of the Problem

To date, a large number of high temperature plastics with excellent mechanical properties have been developed. Their loading with high-modulus fibers results in further efficient enforcement of the composites. However, they can hardly be employed for anti-friction applications, both because of high friction coefficients and (micro-abrasive) damaging effects on (metal) counterparts. This problem can be solved by adding solid lubricant fillers [17,18]. Most often, polytetrafluoroethylene (PTFE; IUPAC name: poly (1,1,2,2-tetrafluoroethylene)) is used to reduce wear rate of polyetheretherketone (PEEK) by several orders of magnitude [19,20,21,22,23]. However, at PTFE loading mechanical properties of composites are significantly reduced. Simultaneous addition of several fillers compensates for this and provides the specified high properties. However, the number of their combinations can be quite large. In this regard, a scientifically based approach is required to select the optimal composition of a composite using a limited set of experimental data.

Polyphenylene sulfide (PPS) is a high temperature structural thermoplastic having high operational properties: process ability; stiffness; heat-, impact-, deformation-, and chemical resistance; as well as high adhesion to a large number of materials (glass, ceramics, titanium, bronze, and steel). This determines the scope of its application: electrical, mechanical engineering, aircraft and automotive, as well as oil and gas industries, etc. However, neat PPS possesses low wear resistance and high friction coefficient (>0.35). This drawback limits its use in friction units [24,25].

Simultaneous addition of solid lubricant and reinforcing fillers in polymer matrixes increases their mechanical and tribological characteristics. This expands potential application areas of polymer composites having specified operational properties in various friction units.

PEEK (IUPAC name: poly (oxy-1,4-pheleneoxy-1,4-phenylene-carbonyl-1,4-phenylene)) is an expensive high-strength, high-temperature advanced thermoplastic polymer widely used in high technologies. It possesses a unique combination of operational properties: high strength and toughness, thermal and chemical resistance, as well as biocompatibility. By varying composition of fillers one can change its characteristics and expands application areas. Reinforcing fibers (carbon, glass, aramid fibers, etc.) are usually used to increase mechanical properties of the PEEK- and PPS-based composites [26,27].

Designing of new materials by experimental methods in laboratories is based on previous data in most cases. Meanwhile, a time lapse required to create a prototype is much less than to study its properties. This is due to necessity of establishing a large number of characteristics: deformation-strength, thermophysical, tribological, electrophysical, technological, etc. A set of the most important properties depends on application areas of the composite material. The quantification of these characteristics by experimental methods requires significant material and time costs. However, these results are reliable.

Application of theoretical research methods (including methods of computational mechanics) demands for development of physical and mathematical models, as well as their implementation [28]. Different structural features should be used in the model depending on its complexity.

Initial data are reference points of dependences of material operational properties versus values of control parameters (regardless of whether they have been obtained by computational or physical experiments). The main idea of the implemented method of computer-aided design of materials is as follows. A 3D surface of each dependence between the operational properties of the material and the control parameters is drawn on the basis of the reference points [15,16]. A region is formed where contours corresponding to the given limits on the material properties are allocated on the plane of the control parameters. Within this region, the control parameters ensure that values correspond to a specific operational property of the material in the given range. Combination of all obtained regions determines a volumetric area of their intersection. The values of all operational properties correspond to the prescribed limits inside it. Coordinates define the specified values of the control parameters. In advance, the control parameters are normalized from 0 to 1 to combine the areas by coordinates.

Implementation of this approach is relevant for design processes of multicomponent composites that should simultaneously possess a complex of high mechanical and tribological properties. Often this can be only achieved in mutually exclusive ways. However, in the proposed approach these limits are taken into account. Their unattainable combinations should be ignored in some cases. Sometimes, it is necessary to reduce the severity of the requirements for one of them. 

High-strength, high-temperature PEEK and PPS thermoplastics are located at the top of the pyramid of structural plastics. This fact, as well as high chemical stability and technological efficiency, have caused their widespread use in high-tech industries, primarily aerospace. However, the analysis of the mechanical properties (including elastic modulus and yield strength) is necessary (as for all structural materials) in order to assess their application in heavily loaded friction units such as seals, thrust bearings and other parts of pumps, centrifuges, and motors. These parts are often operated in a lubricationless or aggressive environment at high temperatures. In turn, high loads and temperatures (although, much lower than melting points) in the friction units significantly accelerate the oxidation processes in PEEK and PPS. This intensifies their wear rate to a significant extent. Thus, the antifriction polymer composites based on the high-strength, high-performance plastics are in demand by the industry. However, the challenges of designing the composites are associated not only with the technological aspects of their formation, but also with the study of the effect of the filler types and quantity on their mechanical properties and wear patterns. The motivation for this study has been largely determined by such a statement of the issue.

This paper is aimed at composition design and justification of the high-strength wear-resistant PEEK- and PPS-based composites. Their operational properties were specified in the form of bilateral and unilateral limits. A database of experimental research results was defined and relationship between the composite compositions and their operational properties was identified. This information was the basis for solving the problem of designing the materials having specified properties. In some cases, the initial data were obtained theoretically using numerical methods of computational mechanics, but the methodology was generally the same.

## 3. Materials and Methods

The PEEK- and PPS-based composites were investigated to achieve a set of mechanical and tribological properties by simultaneously filling with short carbon reinforcing (micro) fibers and PTFE solid lubricant particles.

The “Victrex” PEEK powder (450 PF, Victrex plc, Lancashire, UK) with an average particle size of 50 μm, the “Ticona Fortron” (0205B4, Celanese Corporation, Irving, TX, USA) PPS powder with an average particle size of 20 μm, fillers—PTFE polytetrafluoroethylene (particle size of 6, …, 20 μm, F4-PN20 grade, “Ruflon” LLC, Perm, Russia)—and carbon fibers (CF) with a length of 200 μm and an average diameter of 7.5 μm (“ZUKM” LLC, Chelyabinsk, Russia) were used in this studies.

The PEEK- and PPS-based composites were obtained by hot pressing at a specific pressure of 14 MPa and a temperature of 400 °C and 320 °C, respectively. Subsequent cooling rate was 2 °C/min. The polymer binder powders and the fillers were mixed by dispersing the suspension components in alcohol using a “PSB-Gals 1335-05” ultrasonic cleaner (“PSB-Gals” Ultrasonic equipment center, Moscow, Russia). Processing time was 3 min; generator frequency was 22 kHz. After mixing, a suspension of the components was dried in an oven with forced ventilation for 3 h at a temperature of 120 °C. The use of alcohol as a mixing medium suggested the absence of volatiles in the ready-made mixtures for hot pressing.

Shore D hardness was determined using an “Instron 902” facility (Instron, Norwood, MA, USA) in accordance with ASTM D 2240 [29].

Tensile properties of the PEEK-based samples were measured using an “Instron 5582” electromechanical testing machine (Instron, Norwood, MA, USA). Shape of the samples met the requirements of standard ASTM D 638 [30]. Due to low ductility of the PPS-based composite samples, their mechanical properties were determined by three-point bending tests using the same “Instron 5582” machine in accordance with standard ASTM D 790 [31] (size of the test specimens was 3.5 × 10 × 70 mm).

“Ball-on-disk” dry friction wear tests of the PPS- and PEEK-based composites were performed using a “CSEM CH-2000” tribometer (CSEM, Neuchâtel, Switzerland) in accordance with ASTM G99 [32] (load was 10 N; sliding speed was 0.3 m/s). Two ball-shaped counterparts 6 mm in diameter were made of GCr15 bearing steel and Al_2_O_3_ ceramics (distance was 3 km; radius of the rotation trajectory was 10 mm; rotation speed was 286 rpm).

An “Neophot 2” optical microscope (Carl Zeiss, Oberkochen, Germany) was used to examine wear track surfaces after testing. The permolecular structure of the composites was studied on the cleaved surfaces of the notched specimens mechanically fractured after exposure in liquid nitrogen. A “LEO EVO 50” scanning electron microscope (Carl Zeiss, Oberkochen, Germany) was used (accelerating voltage was 20 kV).

## 4. Results and Discussion

### 4.1. Design of PEEK-Based Wear-Resistant Composites

Initially, 3D dependences of operational properties of the PEEK-based composites containing 5%, 10%, and 17% CF, as well as 5%, 10%, and 17% PTFE (hereinafter all percentages are presented by weight) on control parameters were drawn (Figure 1 and Figure 2). Experimental data supplemented to continuous functions using the Lagrange interpolation polynomial were used. Then, regions corresponding to their specified operational properties (Table 1) were limited on the surfaces.

Note that variation of fillers content did not always caused a change in the values of the operational properties (Figure 1). This fact complicated determination of the composite compositions. The marked limits of each material characteristic met the above requirements. The “0” coordinate corresponded to the minimum degree of filling (5%), whereas the “1” corresponded to the maximum (17%).

Overlapping of these regions defined one common area (the green one in Figure 2). The control parameters within this area corresponded to the values of the operational properties within the prescribed ranges. It is seen that use the material composition corresponding to the center of the green area in Figure 2 (for example, PEEK + 13.4% CF + 9.8% PTFE) is preferable. This ensured that the values were within the specified range, even taking into account possible deviations from technological parameters during manufacturing of the composites or dispersion of the properties of the matrix and the fillers.

To illustrate the proposed methodology, the mechanical and tribological characteristics of the PEEK-based composites of the optimal composition providing the specified properties are shown in Table 2 and Table 3. Both two- and three-component composites contained 10% fillers which matched the range of the values can be determined according to Figure 2.

Note that the proposed methodology for designing the three-component polymer composites is based on the search for their quantity ranges, but not identification of strictly specified contents of each filler. For this reason, the data in Table 2 and Table 3 show that the specified operational properties will be achieved at the center of this range. In this study, it has luckily coincided that the pre-tested samples (compositions) correspond to the center of the range. As a rule, a completely different composition is designed based on precalculated data and used to make samples then [16].

SEM micrographs of the permolecular structure of unfilled PEEK sample, as well as the PEEK-based two- and three-component composite samples are shown in Figure 3. It is seen that loading of PTFE significantly worsened uniformity of the structure (Figure 3c,d). However, satisfactory interfacial adhesion between CF and the PEEK matrix was evident (Figure 3e,f). A quasi-uniform fiber distribution was found in the polymer matrix of the three-component composites (Figure 3g,h), but the structure was not uniform in this case.

Unfortunately, the authors do not have Computed Tomography data to evaluate the volumetric distribution of the filler in the polymer matrix. However, the composites have been made from a powder mixture contained milled fibers with a length of 200 μm and an aspect ratio of less than 10. Authors faced no difficulties in ensuring their uniform distribution and eliminating agglomeration in the polymer matrix. The low-magnification SEM micrographs at Figure 3 evidence for the fairly uniform distribution of the fillers.

Diagrams of the volumetric loss of the PEEK-based composites are presented in Figure 4. It is seen that the maximum values corresponded to the three-component composites. The “PEEK + PTFE” two- and “PEEK + PTFE + CF” three-component composites possessed the smallest wear regardless of the counterpart type. Nevertheless, wear of the “PEEK + CF” composite sample on the metal counterpart was lower than that of the unfilled PEEK sample. At the same time, it was about an order of magnitude higher on the ceramic counterpart.

An analysis of the wear track surfaces of the samples was done to identify reasons of the observed phenomena. Optical photographs of the surfaces of the unfilled PEEK sample and the PEEK-based composite samples, as well as the metal and ceramic counterparts, are shown in Figure 5. Microgrooves oriented in the sliding direction had been formed on the surface of the unfilled PEEK sample when tested on the metal counterpart (Figure 5a). Signs of wear are evident on the surface of the metal counterpart. This indicates that the nature and degree of wear of the unfilled PEEK sample was determined by the microcutting effect of the partly worn out metal counterpart and, probably, wear debris (Figure 5b). A low level of the polymer sample wear was found after sliding on the ceramic counterpart in comparison with the previous case. Wear scars were smaller on the ceramic counterpart surface (Figure 5d). However, there were both separate shallow longitudinal microgrooves and a material transfer film visualized through rainbow colors. Again, note that the polymer sample wear decreased by approximately four times in this case compared with the metal–polymer friction pair (Table 3).

Both polymer and counterpart surfaces were least worn after testing the “PEEK + 10% PTFE” composite sample (Figure 5f,h). PTFE particles were quasi-uniformly distributed in the form of rather large inclusions as was seen on the polymer composite surface (Figure 5e). A thin transfer film was evident on the surfaces of the counterparts (Figure 5f,h). According to the authors, it had protected the surfaces of the counterparts and the polymer samples from wearing by micro-abrasive wear debris. When slid on the metal counterpart, CF being protruded above the surfaces of the reinforced composite samples had reduced material wear compared to the unfilled PEEK samples due to higher hardness (Table 3). At the same time, they had exerted a cutting effect on the metal counterpart surface caused its heavy wear (Figure 5k). Wear of the ceramic counterpart surface by CF was minimal (Figure 5m). On the other hand, the harder ceramic counterpart had caused impact wear of the friction surfaces of the composite samples due to its hardening and increasing brittleness. As a result, its wear resistance had been significantly reduced (Table 3). This had been accompanied by intensive accumulation of wear debris on the wear track surface (Figure 5l).

Wear of both counterparts was almost not found after tests of the “PEEK + 10% PTFE + 10% CF” three-component composite sample (Figure 5o,q). The friction surface of the polymer composite was smooth with almost no grooves (Figure 5n,p). Its roughness was lower than that on the unfilled PEEK sample. Thus, it can be concluded that PTFE had acted as a solid lubricant under conditions of dry sliding friction. This had enabled to protect the surfaces of the sample and the counterparts from the micro-abrasive damaging. Friction coefficient and wear rate had significantly decreased due to this fact (Table 3).

In so doing, the results of the wear track surface analysis of both composites and counterparts made it possible to conclude that the “PEEK + 10% PTFE + 10% CF” three-component composite was the best among the investigated ones. This corresponded to the area of the specified operational properties and the compositions determined by the proposed methodology. It can be recommended to use in both metal–polymer and ceramic–polymer friction units. In the first case, wear resistance increased by fourteen times. In the second one, it was improved eight-fold, whereas the elastic modulus increased by 1.5-fold. The amount of the CF can be increased up to 14, …, 15% to ensure the characteristics fall within the prescribed limits (the green area in Figure 2).

### 4.2. Design of PPS-Based Wear-Resistant Composites

Another high-temperature thermoplastic polymer (PPS) having a matrix similar in structure and some properties were studies in a similar way. However, due to difference in the tribological and mechanical properties the prescribed limits and the contents of the CF reinforcing and PTFE solid lubricant fillers were increased. In particular, based on published data [33,34] and the results of preliminary studies of the authors, the structure and the properties of the PPS-based composites were modified by loading 15, 22.5, and 30% CF, as well as 10, 15, and 20% PTFE. Then, the results were treated according to the methodology described in the previous section.

Initially, the specified operational properties for the PPS-based composites were prescribed (Table 4). The 3D surfaces and their corresponding contour fields for the PPS-based composite had different degrees of filling with CF and PTFE, see Figure 6 (PPS content is on the *X*-axis, CF content is on the *Y*-axis). The degrees of filling were the control parameters. The obtained results have proven the influence of these parameters on such operational properties of the composites as (i) density, (ii) bending modulus and strain, (iii) flexural strength, (iv) Shore D hardness, (v) friction coefficients, and (vi) volumetric wear on the metal and ceramic counterparts.

A region of the specified operational properties is filled in color in Figure 6a. Some of these limits were inactive (for example, density). In particular, the lower limit for bending modulus is less than 8000 MPa. These values were found empirically using the published data on the properties of the PPS-based composites. The limits were unilateral in the form of “no more” or “no less”. As a result, the region was formed on the plane of the control parameters that ensured the fulfillment of these limits. When the values of the control parameters corresponded to this region, it is provided that the operational properties were within the given range. To take all of them into account, the curves were matched and an intersection area of the prescribed limits was obtained (the green area in Figure 7). Then, these data on the PPS-based three-component composites were analyzed. The data processing methodology was the same as for the PEEK-based composites (see previous chapter). The best solution was also in the center of the green area. It corresponded to the “PPS + 25.5% CF + 16% PTFE” composition. Content of the CF was approximately doubled, while PTFE increased one and a half times compared with the PEEK-based composites.

To once again illustrate the advantages of the proposed methodology, the mechanical and tribological characteristics of the PPS-based composites of the optimal composition according to the specified properties are shown in Table 5 and Table 6.

As the two-component composites containing 15% PTFE were used for comparison, the three-component composites were filled with the same amount of fluoroplastic. Content of the CF was 22.5% for both two- and three-component composites. This corresponded to the calculated composition and was in the specified range (the green area in Figure 7).

SEM micrographs of the permolecular structure of unfilled PPS samples, as well as the PPS-based two- and three-component composites are shown in Figure 8. On the one hand, loading of PTFE worsened significantly the structure uniformity of the composites (Figure 8c,d). On the other hand, interfacial adhesion between the CF and the PPS-matrix was weaker compared to the PEEK-matrix (Figure 8e,f). The permolecular structure of the three-component composites had a quasi-uniform fiber distribution in the polymer matrix (Figure 8g,h).

Much like the data on PEEK-permolecular structure, the low-magnification SEM-micrographs shown in Figure 8 are given to prove the fairly uniform filler distribution in the PPS matrix.

Determined tribological properties are graphically illustrated in Figure 9. One can conclude about the advantages of the PPS-based three-component composites. Samples of both “PPS + PTFE” two- and “PPS + PTFE + CF” three-component composites had minimal wear regardless of the counterpart type. However, wear of the “PPS + CF” composite sample on the metal counterpart was less than that of the unfilled PPS sample. On the other hand, it is approximately an order of magnitude higher on the ceramic counterpart. It should be noted that PTFE in the three-component composite showed its solid–lubricant properties to various degrees when tested on the metal and ceramic counterparts (Figure 9, columns 4).

Wear track surfaces of the friction pairs were then analyzed to determine reasons of the observed phenomena; these phenomena occur because of optical photographs of the sample wear track surfaces of unfilled PPS and the PPS-based composites, as well as the metal and ceramic counterparts are presented in Figure 10. It is seen that microgrooves oriented in the sliding direction had formed on the surface of the unfilled PPS samples during the test on the metal counterpart (Figure 5a). The same microgrooves were found on the surface of the metal counterpart (Figure 10b). At the same time, the surface of the ceramic counterpart was slightly damaged. Its worn out area (scar) was smaller than the similar one on the surface of the metal counterpart (Figure 10d). However, surface roughness of the wear track of unfilled PPS sample was less than after testing on the metal counterpart.

It is seen that the counterparts were nearly worn out during the tribological tests of the “PPS + 15% PTFE” composite sample (Figure 10f,h). A thin transfer film had been formed on the metal and ceramic counterpart surfaces. It is suggested that it had protected the composite samples and the counterparts from micro-abrasive wearing. These results correlated with the data on the volumetric wear rate depending on the type of filler (Figure 9). The transfer film had formed with PTFE on the metal and ceramic counterpart surfaces (Figure 10e–h) and increased wear resistance of the composite samples by 8, …, 10 times in both cases. However, loading of the CF resulted in abrasive wear on the surface of the metal counterpart. In addition to deep wear scar, microgrooves and scratches on the metal friction surface were evident (Figure 10k). Such heavy wear of the surface was not observed after sliding on the ceramic counterpart (Figure 10m). Moreover, no longitudinal grooves were found on the surfaces of the polymer composites (Figure 10l). Therefore, a multiple increase in wear of the PPS-based composites enforced with the CF can be attributed to damaging impact of the ceramic counterpart. Its cyclic compressive–shear action on the reinforced composite samples had caused failure by the fatigue mechanism. The same wear scar was formed on the surface of the metal counterpart after testing the “PPS + 15% PTFE + 22.5% CF” three-component composite samples (Figure 10o) similarly to the “PPS + 20% CF” two-component composite ones. In this case, CF protruded above the wear track surface of the polymer composite samples.

According to the authors, the fibers had exerted abrasive effect on the metal counterpart, acted above the wear surface under conditions of low adhesion of CF with the polymer matrix. However, CF cutting effect on the surface was not found after sliding on the harder ceramic counterpart (Figure 10q). In so doing, the PTFE acted as a solid lubricant protecting the polymer sample surface. As compared with sliding on the metal counterpart, the friction coefficient and wear had decreased by an order of magnitude (Figure 9).

Thus, the result analysis on the mechanical and tribological properties, as well as the wear track surfaces of the studied samples, indicated the advantage of the “PPS + 15% PTFE + 22.5% CF” three-component composite. This corresponded to the specified values of the operational properties according to the composition determined using the described above method. Unlike the “PEEK + 10% PTFE + 10% CF”, the “PPS + 15% PTFE + 22.5% CF” composite was more efficient in terms of mechanical and tribological properties when used in ceramic–polymer friction units.

The results of a comparison of the three-component composites based on both polymer matrixes with an almost identical degree of filling are presented in Table 7. Note that these compositions were not optimal for the PEEK-based composites due to the content of both fillers. 

It is seen that they were one and a half, and two times higher than the level determined using the Figure 2. The most highly filled “PEEK + 15% PTFE + 20% CF” composite had enhanced tribological properties compared to the PPS-based composites with the same degree of filling. However, they were lower than that of the “PEEK + 10% PTFE + 10% CF” composite determined by the above methodology. This once again allows us to conclude about its efficiency and sensitivity to the features of the original polymer matrix (in particular, the permolecular structure and adhesion of the matrix to fillers). These features and their influence on the experimental results are illustrated in Figure 11 and Figure 12. Thus, optical photographs of the sample wear surfaces of the PEEK- and PPS-based three-component composite samples of the identical composition, as well as the metal and ceramic counterparts, are shown in Figure 12. In contrast to the PPS-based composites, when wear of the metal counterpart was high, cutting effect of the PEEK-based composites was minimal (Figure 12b). Also, volumetric wear of the PEEK-based composite sample was significantly less than that of the PPS-based one. We suggest that this resulted from high adhesion of the CF to the PEEK-matrix (Figure 11b) in combination with providing PTFE solid lubricant effect.

Ultimately, the results of the tribological tests, analysis of the wear track surfaces, and the permolecular structure have proven the efficiency of the designed composites, whose compositions had been determined with the use the proposed methodology.

By the way of summarizing the above described structure, mechanical, and tribological properties of PEEK- and PPS-based composites the authors want to stress the following. The polymer composites having excellent tribological properties are widely used in various transport systems due to their high self-lubricating ability and temperature stability. To achieve the required operational properties, various types of fillers are loaded into the polymer matrix (that is defined by the functional tasks to be solved). Thus, reinforcing fibers (carbon, glass, aramid, etc.) as well as solid lubricant fillers (primarily PTFE) are usually used to simultaneously improve the mechanical and tribological properties of the PEEK- and PPS-based composites. In doing so, the fibers improve strength of the composites through increasing the elastic modulus, whereas solid lubricant particles reduce the friction coefficient and wear rate [35]. V. Rodriguez et al. [36] have shown that loading of PEEK with graphite and PTFE in a content of 10 wt. % enabled to simultaneously improve strength (elastic modulus increases by 4.6 times) and tribological properties (the friction coefficient decreases from 0.40 to 0.28). Zang Z. et al. [37] tested the epoxy-based composite loaded with SCF, graphite, PTFE, and nano-TiO_2_. The best wear resistant composition was achieved in the composite with 15 vol. % graphite + 5vol. % nano-TiO_2_ + 15 vol. % SCF exhibits a specific wear rate of 3.2 × 10^−7^ mm^3^/Nm, which is ~100 times lower when compared to the neat epoxy. Zang et al. [38] tested the PPS-based composites loaded with SCF and graphite when friction has been under diesel lubrication conditions. Loading with the fillers has significantly reduced the friction coefficient and wear rate. Thus, published data usually describe the results of an increase in the elastic modulus (from tens of percent to units of times), as well as a decrease in the friction coefficient and wear rate (also up to several times). The results obtained in the current study are quite competitive with the data from the literature.

Also, in this paper, two issues have been solved: First, the optimal contents of two fillers have been determined (each of the fillers ensured the achievement of the opposite functional requirements). Second, the wear patterns of the three-component polymer composites under conditions of metal–polymer and ceramic–polymer tribological contacts have been analyzed. In doing so, the reasons for improving wear resistance of both the polymer composites and the harder counterparts have been shown.

The idea of the manuscript has been formulated to a greater extent as development of a methodology to design the polymer composites having both high strength and wear resistance. These two properties have been often achieved in opposite ways. For this purpose, the authors have not used any “exclusive” fillers, primarily to illustrate the possibility of the developed methodology. However, this does not exclude options to design some composites based on advanced polymer matrixes, compatibilizers, reinforcing inclusions, or (nano) modifiers. In addition to mechanical and tribological properties, the composites can be designed in accordance with the requirements of biocompatibility, electrical or thermal conductivity, etc.

## 5. Conclusions

The material design methodology for the composites with specified properties presented in this paper is of clarity and relatively simple in implementation. Its use requires a relatively small amount of the initial data obtained on the basis of the results of the computational or physical experiments. The methodology enabled determining the range of valid values of the control parameters and choosing their most acceptable combination. Moreover, implementation of the methodology made it possible to avoid obviously unattainable combinations of the material quality requirements.

The optimal degrees of filling of composites based on polyphenylene sulfide and polyetheretherketone with carbon microfibers and polytetrafluoroethylene particles were determined. For this purpose the results of the physical experiments and the described methodology were used in order to obtain the specified mechanical and tribological properties. The PEEK-based composites with the same degree of filling with the reinforcing fibers and the solid lubricant particles exerted a less damaging effect on the metal counterpart. This was most likely due to more uniform permolecular structure and greater elasticity of the matrix compared to PPS, since elongation to failure of the unfilled polymers differed by about five times.

The proposed methodology for designing multicomponent composites is versatile and can be used to design various composites. Its implementation does not impose any limits on the specified properties of the material matrix or the reinforcing inclusions. The initial data on the operational characteristics can be obtained experimentally or numerically. The methodology enables the design the high-strength wear resistant composites, which are well suited for using in the metal–polymer and ceramic–polymer friction units.

## Figures and Tables

**Figure 1 materials-13-00524-f001:**
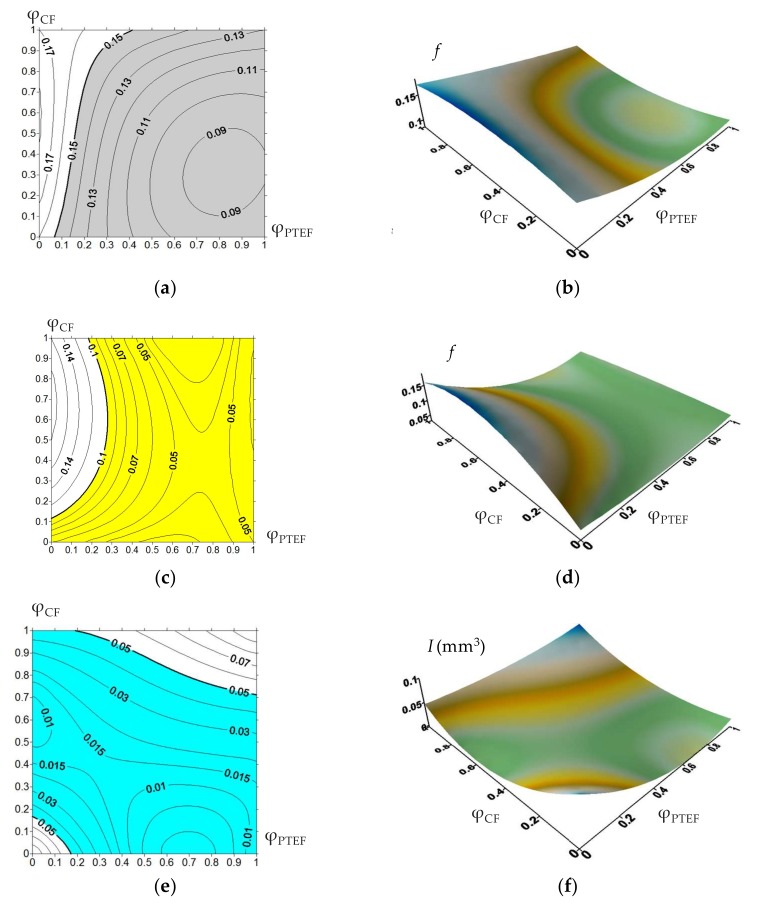
Operational properties of the PEEK-based composites vs. degree of CF and PTFE filling: friction coefficient on metal counterpart (**a**,**b**), friction coefficient on ceramic counterpart (**c**,**d**), volumetric wear on metal counterpart (**e**,**f**), volumetric wear on ceramic counterpart (**g**,**h**), elastic modulus (**i**,**j**), tensile strength (**k**,**l**), elongation (**m**,**n**), and Shore D hardness (**o**,**p**).

**Figure 2 materials-13-00524-f002:**
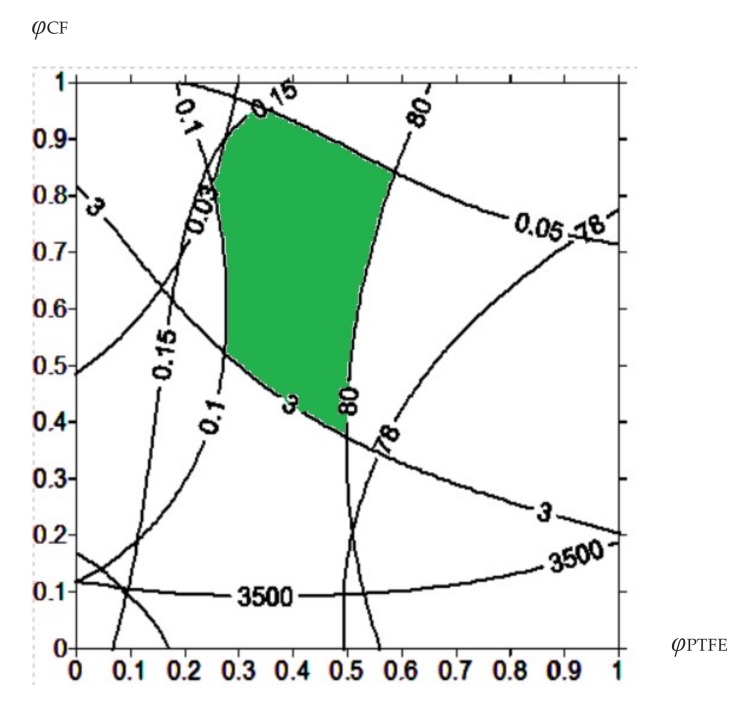
Diagram of the control parameters to ensure that the tribological properties meet the specified limits for the PEEK-based composites.

**Figure 3 materials-13-00524-f003:**
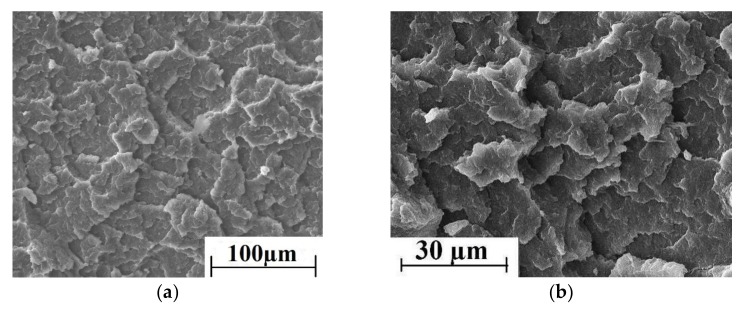
SEM micrographs of the permolecular structure: (**a**,**b**) PEEK and (**b**–**h**) the PEEK-based composites ((**c,d**) PEEK + 10% PTFE, (**e,f**)–PEEK + 10% CF, and (**g,h**) PEEK + 10% PTFE + 10% CF.)

**Figure 4 materials-13-00524-f004:**
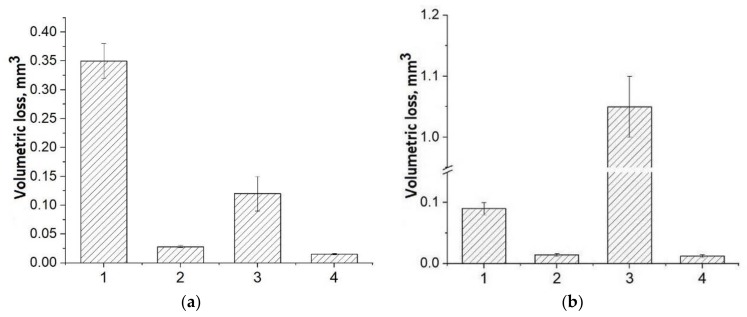
Volumetric wear: On (**a**) the metal counterpart and (**b**) on the ceramic counterpart (1–PEEK, 2–PEEK + 10% PTFE, 3–PEEK +10% CF, 4–PEEK +10% PTFE + 10% CF).

**Figure 5 materials-13-00524-f005:**
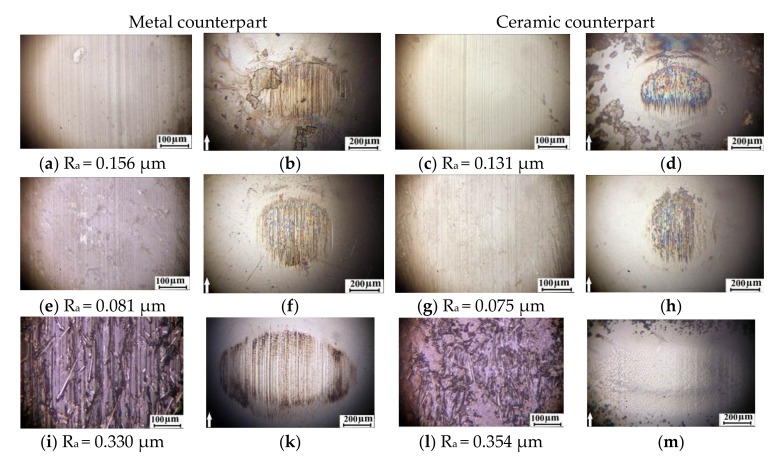
Optical images of the wear track surfaces and the counterparts at the steady state wearing stage: (**a**–**d**) PEEK; (**d**–**q**) the PEEK-based composites ((**e****–h**)–PEEK + 10% PTFE; (**i****–m**)–PEEK +10% CF; (**n****–q**)–PEEK +10% PTFE + 10% CF).

**Figure 6 materials-13-00524-f006:**
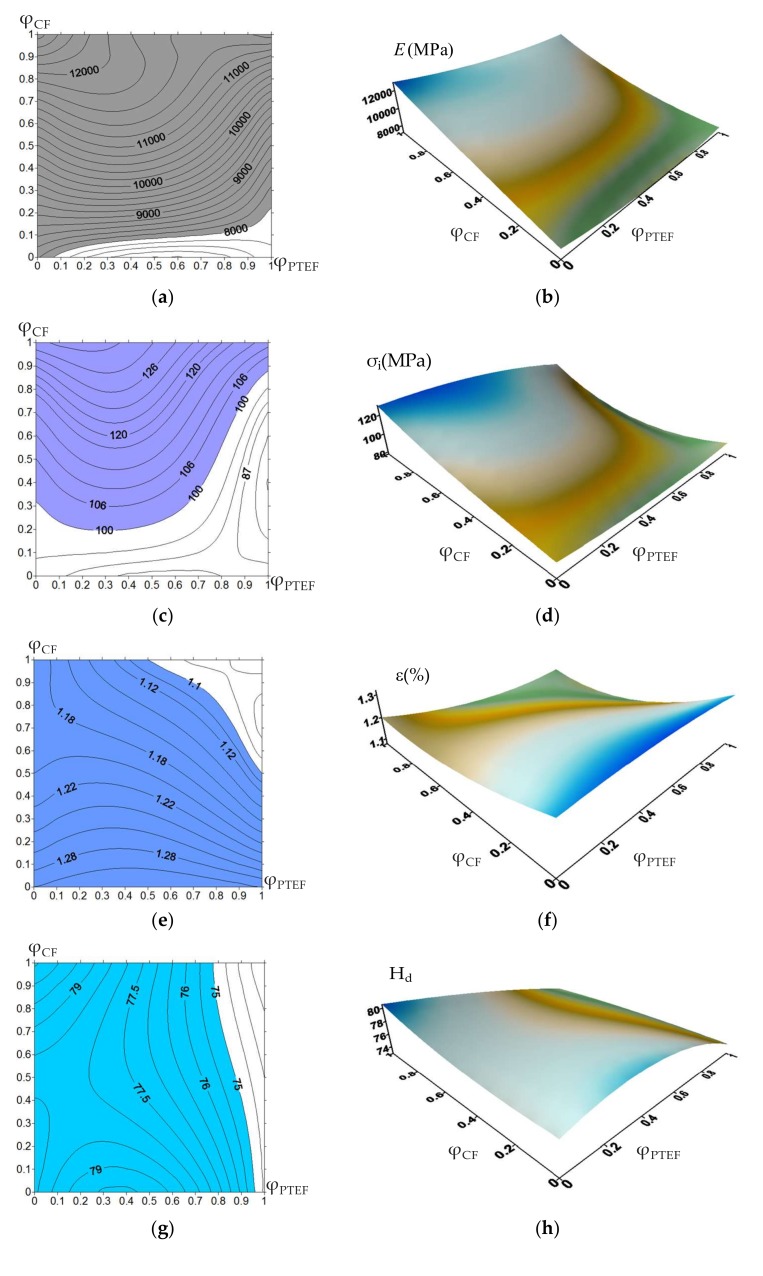
Operational properties of the PPS-based composites vs. degree of CF and PTFE filling: bending modulus (**a**,**b**), flexural strength (**c**,**d**), bending strain (**e**,**f**), Shore D hardness (**g**,**h**), friction coefficient on metal counterpart (**i**,**j**), volumetric wear on metal counterpart (**k**,**l**), friction coefficient on metal counterpart (**m**,**n**), and volumetric wear on metal counterpart (**o**,**p**).

**Figure 7 materials-13-00524-f007:**
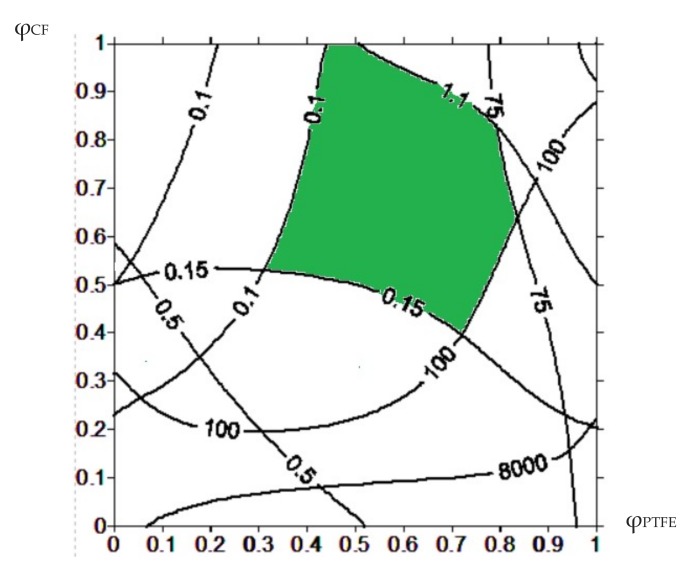
Diagram of the control parameters to ensure that the tribological properties meet the specified limits for the PPS-based composites.

**Figure 8 materials-13-00524-f008:**
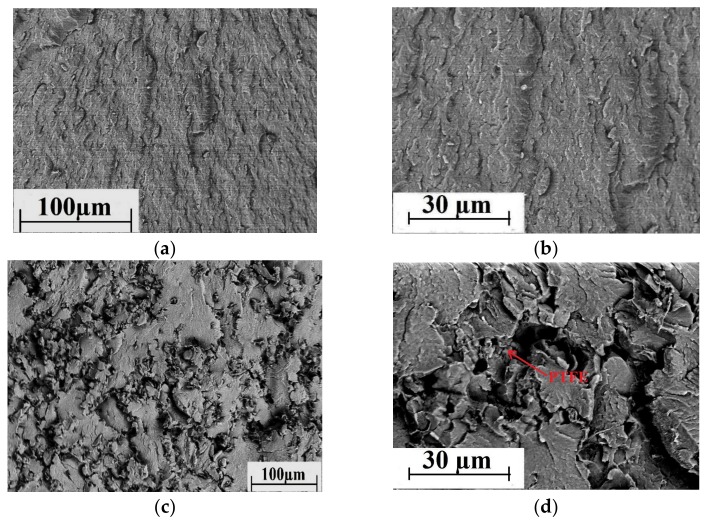
SEM micrographs of the permolecular structure: (**a**,**b**) PPS, (**c**–**h**) the PPS-based composites ((**c,d**) PPS + 15% PTFE, (**e,f**) PPS + 20% CF, (**g,h**) PPS + 15% PTFE + 22.5% CF).

**Figure 9 materials-13-00524-f009:**
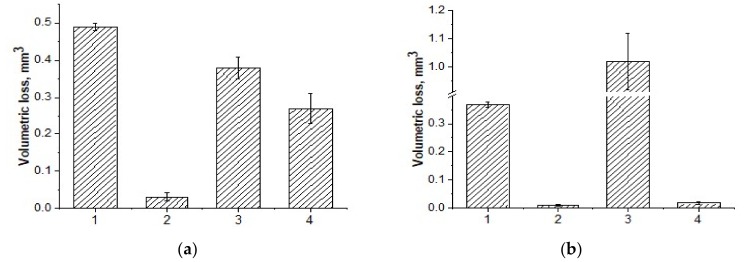
Volumetric wear: (**a**) on the metal counterpart and (**b**) on the ceramic counterpart (1–PPS, 2–PPS + 15% PTFE, 3–PPS + 20% CF, 4–PPS + 15% PTFE + 22.5% CF).

**Figure 10 materials-13-00524-f010:**
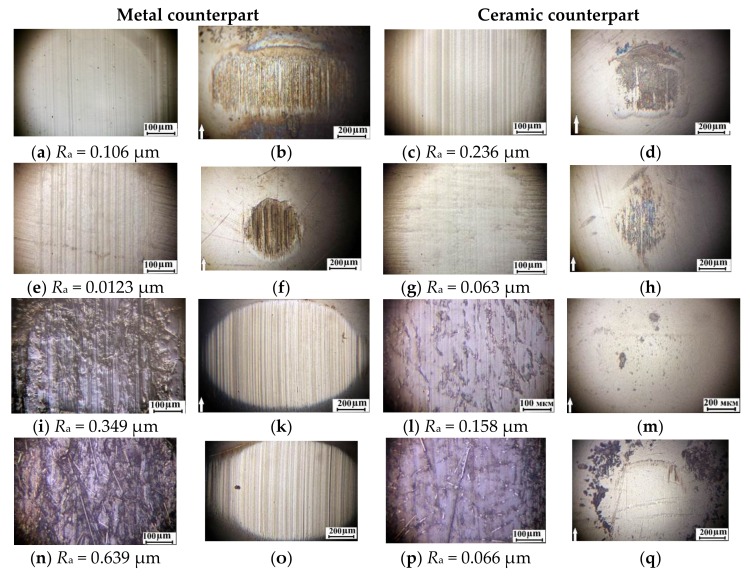
Optical images of the wear track (sample) surfaces and the counterparts at the steady state wearing stage: (**a**–**d**) PPS, (**d**–**q**) PPS-based composites ((**e**–**h**) PPS + 15% PTFE, (**i**–**m**) PPS + 20% CF, (**n**–**q**) PPS + 15% PTFE + 22.5% CF).

**Figure 11 materials-13-00524-f011:**
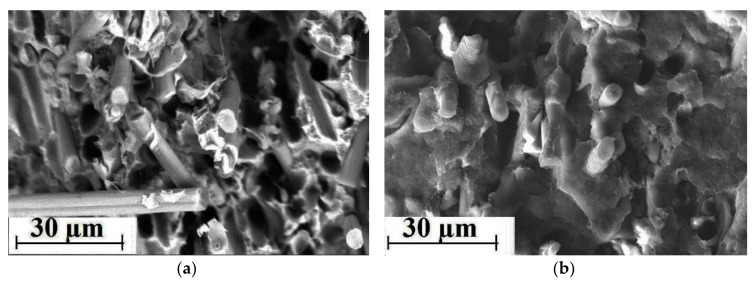
SEM micrographs of the permolecular structure of the three-component composites: (**a**) PPS + 15% PTFE + 22.5% CF and (**b**) PEEK + 15% PTFE + 20% CF.

**Figure 12 materials-13-00524-f012:**
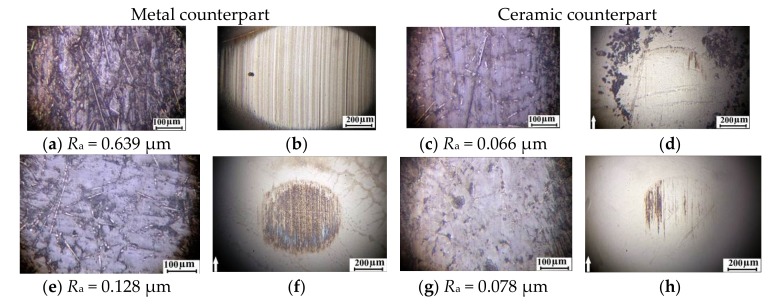
Optical images of the wear track (sample) surfaces and the counterparts at the steady state wearing stage: (**a**–**d**) PPS + 15% PTFE + 22.5% CF and (**e**–**h**) PEEK + 15% PTFE + 20% CF.

**Table 1 materials-13-00524-t001:** Specified operational properties for the PEEK-based composites.

Properties	Values
Shore D hardness	>78
Elastic modulus, Mpa	>3.500
Tensile strength, Mpa	>80
Elongation, %	>3%
Friction coefficient on metal counterpart	<0.15
Friction coefficient on ceramic counterpart	<0.1
Volumetric wear on metal counterpart (distance of 3 km), mm^3^	<0.05
Volumetric wear on ceramic counterpart (distance of 3 km), mm^3^	<0.03

**Table 2 materials-13-00524-t002:** Mechanical properties of the PEEK and PEEK-based composites.

Filler Composition, % (wt.)	Density, *ρ*, g/cm^3^	Shore D Hardness	Elastic Modulus, *G*, MPa	Tensile Strength, σ, MPa	Elongation, ε, %	Crystallinity, µ, %
PEEK	1.308	80.1 ± 1.7	2.840 ± 273	106.9 ± 4.7	25.6 ± 7.2	32.5
+ 10% PTFE	1.325	77.3 ± 0.3	2.620 ± 158	83.9 ± 2.4	5.0 ± 0.8	28.8
+ 10% CF	1.331	82.5 ± 0.7	4.821 ± 70	125.4 ± 0.5	1.5 ± 0.3	26.8
+ 10% PTFE + 10% CF	1.385	80.1 ± 0.5	4.238 ± 125	103.8 ± 5.2	3.9 ± 0.5	22.7

**Table 3 materials-13-00524-t003:** Tribological properties of the PEEK-based composites.

Filler Composition, % (wt.)	Friction Coefficient, ƒ		Volumetric Wear, mm^3^	
	Metal Counterpart	Ceramic Counterpart	Metal Counterpart	Ceramic Counterpart
PEEK	0.34 ± 0.03	0.27 ± 0.02	0.35 ± 0.03	0.09 ± 0.01
+ 10% PTFE	0.17 ± 0.02	0.1 ± 0.02	0.028 ± 0.002	0.014 ± 0.02
+ 10% CF	0.24 ± 0.02	0.25 ± 0.02	0.12 ± 0.03	1.05 ± 0.26
+ 10% PEEK + 10% CF	0.11 ± 0.01	0.07 ± 0.02	0.015 ± 0.001	0.012 ± 0.02

**Table 4 materials-13-00524-t004:** Specified operational properties for the PPS-based composites.

Properties	Values
Shore D hardness	>75
Bending modulus, MPa	>8.0
Flexural strength, MPa	>100
Bending strain, %	>1.1%
Friction coefficient on metal counterpart	<0.15
Friction coefficient on ceramic counterpart	<0.1
Volumetric wear on metal counterpart (distance of 3 km), mm^3^	<0.5
Volumetric wear on ceramic counterpart (distance of 3 km), mm^3^	<0.1

**Table 5 materials-13-00524-t005:** Mechanical properties of PPS, as well as the PPS-based two- and three-component composites.

Filler Composition, % (wt.)	Density, *ρ*, g/cm^3^	Shore D Hardness	Elastic Modulus, *G*, MPa	Tensile Strength, σ, MPa	Elongation, ε, %
PPS	1.331	79.5 ± 0.5	3.930 ± 71	97.8 ± 1.6	2.60 ± 1.4
+ 15% PTFE	1.410	74.1 ± 0.5	3.596 ± 140	46.0 ± 4.9	1.30 ± 0.1
+ 20% CF	1.435	80.9 ± 0.3	7.893 ± 2197	120.1 ± 18.5	1.30 ± 0.2
+ 15% PTFE + 22.5% CF	1.526	77.2 ± 0.5	10.908 ± 985	113.9 ± 14.3	1.20 ± 0.1

**Table 6 materials-13-00524-t006:** Tribological properties of PPS, as well as the PPS-based two- and three-component composites.

Filler Composition, % (wt.)	Friction Coefficient, ƒ		Volumetric Wear, mm^3^	
	Metal Counterpart	Ceramic Counterpart	Metal Counterpart	Metal Counterpart
PPS	0.30 ± 0.02	0.17 ± 0.03	0.49 ± 0.01	0.37 ± 0.01
+ 15% PTFE	0.09 ± 0.02	0.05 ± 0.002	0.031 ± 0.009	0.02 ± 0.01
+ 20% CF	0.31 ± 0.04	0.23 ± 0.03	0.38 ± 0.03	1.02 ± 0.02
+ 15% PTFE + 22.5% CF	0.15 ± 0.02	0.06 ± 0.02	0.27 ± 0.04	0.018 ± 0.005

**Table 7 materials-13-00524-t007:** Tribological properties of the PEEK- and PPS-based three-component composites with identical degree of filling.

Filler Composition, % (wt.)	Friction Coefficient, ƒ		Volumetric Wear, mm^3^	
	Metal SCB	Ceramic SCB	Metal SCB	Metal SCB
PPS + 15% PTFE + 22.5% CF	0.15 ± 0.02	0.06 ± 0.02	0.27 ± 0.04	0.018 ± 0.005
PEEK + 15% PTFE + 20% CF	0.13 ± 0.01	0.06 ± 0.01	0.016 ± 0.003	0.017 ± 0.002
(comparison) PEEK + 10% PTFE + 10% CF	0.11 ± 0.01	0.07 ± 0.02	0.015 ± 0.001	0.012 ± 0.002

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
