# Peer review of "Material Design Methodology for Optimized Wear-Resistant Thermoplastic–Matrix Composites Based on Polyetheretherketone and Polyphenylene Sulfide"

_materials, 2020, doi:10.3390/ma13030524_

Round 1

Reviewer 1 Report

Line 24-30- English makes no sense as to what the authors are trying to communicate

Line 39-40- Check Grammar

Line 89 – “PEEK is a relatively new class of polymers widely used in advanced technologies.”  Peek has been around for a long time Revise perspective here .

Line 139-143- If the mixing was done with ethanol what was the procedure to evaporate or make sure there is no volatiles in the matrix of the composites. Detail the process completely.

Were CT Scans or other NDE method use to confirm the homogeneity of fiber dispersion in matrix

Line 146-151- Why friction test was done as per ASTM and Mechanical test as per Russian GOST STANDARDS? Please make data reporting as per ASTM or per GOST

Why none of the mechanical data for each concentration was tester and reported as distinct data point   though Figure 6 and 8 show material properties

SEM images need to be of higher magnification so the difference in PTF and CF fiber can be seen.

Figure 10 & 12 the image need to be all under the same illumination source currently it looks like it is not

Conclusion- there is no explanation on why someone should care about this improved tribological performance? What is the end purpose the case seems to be in the introduction but it is not explained clearly

Author Response

Dear respectful Reviewer, 

many thanks for all your valuable remarks. All of them are reasonable and make it possible to improve the text substantially. Please find below the exact replies for all your notes. The corresponding corrections were put in the text of the manuscript and marked with cyan.

thank you once again for all the efforts on improving the manuscript,

on behalf of the authors,

Sergey Panin

Line 24-30- English makes no sense as to what the authors are trying to communicate

Thanks a lot for your remark. This part of the manuscript has been rewritten.

The described methodology is versatile and can be used to design various composites. Its implementation does not impose any limits on the specified properties of the material matrix or the reinforcing inclusions. The initial data on the operational characteristics can be obtained experimentally or numerically. The methodology has enabled the design of the high-strength wear-resistant composites which are well suit for using in the metal-polymer and ceramic-polymer friction units.

Line 39-40- Check Grammar

Thanks a lot for your remark. This part of the manuscript has been rewritten.

A significant variety of structures and properties of the composites have caused the development of numerous mathematical models and methodologies for the prediction of their characteristics [1–4].

Line 89 – “PEEK is a relatively new class of polymers widely used in advanced technologies.” Peek has been around for a long time Revise perspective here.

You are absolutely right. This part of the manuscript has been rewritten.

PEEK is an expensive high-strength high-temperature advanced thermoplastic polymer widely used in high technologies.

Line 139-143- If the mixing was done with ethanol what was the procedure to evaporate or make sure there is no volatiles in the matrix of the composites. Detail the process completely.

Thanks a lot for your remark.

After mixing, a suspension of the components in ethanol was dried in an oven with forced ventilation for 3 hours at a temperature of 120 °C. The use of alcohol as a mixing medium suggested the absence of volatiles in the ready-made mixtures for hot pressing.

Were CT Scans or other NDE methods use to confirm the homogeneity of fiber dispersion in the matrix.

Thanks a lot for your remark.

Unfortunately, the authors do not have Computed Tomography data to evaluate the distribution of the filler in the polymer matrix. However, the composites were made from a powder mixture contained milled fibers with a length of 200 μm and an aspect ratio of less than 10. There were no difficulties in ensuring their uniform distribution and eliminating agglomeration in the polymer matrix. The authors presented SEM micrographs, according to which the distribution of the filler was fairly uniform.

Line 146-151- Why friction test was done as per ASTM and Mechanical test as per Russian GOST STANDARDS? Please make data reporting as per ASTM or per GOST

Thanks a lot for your remark. This part of the manuscript has been rewritten.

The tensile properties of the PEEK-based samples were measured using an “Instron 5582” electromechanical testing machine. The shape of the samples met the requirements of the standard ASTM D 638. Due to low ductility of the PPS-based composite samples, their mechanical properties were determined by three-point bending tests using the “Instron 5582” machine in accordance with the standard ASTM D 790 (size of the test specimens was 3.5×10×70 mm).

Why none of the mechanical data for each concentration was tested and reported as distinct data point though Figure 6 and 8 show material properties

Thanks a lot for your remark.

The authors have much more data on the mechanical properties of the composites. However, it has not been considered necessary to present them all because their volume is too large and they can divert attention from the already given sampled data. According to the authors, it is more important to show the final processed results without distracting the reader to raw data.

SEM images need to be of higher magnification so the difference in PTF and CF fiber can be seen.

Thanks a lot for your remark.

The SEM micrographs were obtained at a magnification of ´3000. A further increase in magnification did not enable us to have contrast images of the permolecular structure of the polymer composites. Accepting this remark of the reviewer, the authors have schematically indicated the PTFE particles and the carbon fibers in the micrographs for clarity.

Figure 10 & 12 the image need to be all under the same illumination source currently it looks like it is not

Thanks a lot for your remark.

Unfortunately, the technique of photographing the surfaces of the wear tracks required to take out the samples and a holder from the testing machine and put them into the optical microscope. In some cases, the lighting conditions were slightly changed during installation. However, it is not possible to fix this problem now because the experiments have been completed.

Conclusion- there is no explanation of why someone should care about this improved tribological performance? What is the end purpose the case seems to be in the introduction but it is not explained clearly

Thanks a lot for your remark. This part of the manuscript has been rewritten.

High-strength high-temperature PEEK and PPS thermoplastics are located at the top of the pyramid of structural plastics [https://www.performanceplastics.com/materials/]. This fact, as well as high chemical stability and technological efficiency, have caused their widespread use in high-tech industries, primarily aerospace. However, the analysis of the mechanical properties (including elastic modulus and yield strength) is necessary (as for all structural materials) in order to assess the possibility of their implementation in heavily loaded friction units such as seals, thrust bearings and other parts of pumps, centrifuges, and motors. These parts are often operated without lubrication in aggressive environments at high temperatures. In turn, high loads and temperatures (although, much lower than melting points) in the friction units significantly accelerate the oxidation processes in PEEK and PPS, intensifying their wear rate to a significant extent. Thus, the antifriction polymer composites based on the high-strength high-performance plastics are in demand by the industry. However, the challenges of designing the composites are associated not only with the technical aspects of their formation but also with the study of the effect of the types and quantity of the fillers on their mechanical properties and wear patterns. The motivation for this study has been largely determined by such a statement of the issue.

Reviewer 2 Report

The authors presented an interesting work on wear resistant composites. The quality needs to be improved before consideration of publication:

(1) the sections need to be numbered correctly

(2) there are too many plots in some of the figures. It would be helpful if the authors can present them more clearly.

Author Response

Dear respectful Reviewer, 

Many thanks for all your valuable remarks. All of them are reasonable and make it possible to improve the text substantially. Please find below the exact replies for all your notes. The corresponding corrections were put in the text of the manuscript and marked with cyan.

Thank you once again for all the efforts on improving the manuscript,

Sergey Panin

The authors presented interesting work on wear-resistant composites. The quality needs to be improved before consideration of publication:

(1) the sections need to be numbered correctly

Thanks a lot for your remark. The numbering of the sections of the manuscript has been corrected.

 (2) there are too many plots in some of the figures. It would be helpful if the authors can present them more clearly.

Thanks a lot for your remark. The plots have been presented with a larger scale in separate figures.

Reviewer 3 Report

The article is interesting and concerns a new group of composite materials.Simulation and design are a very important factor to reduce material selection time. The authors described the results of their latest research in the publication.
Authors should check the correctness of the chemical nomenclature as agreed by UPAC.
The entire introduction concerns methods and problems related to simulation and there is nothing about what these simulations have helped to achieve in the field of modern polymer composites.

Table 1 gives the ranges of properties of composites, but it should be specified which indicators for materials with different fillings were concisely obtained, taking into account standard deviations, so that it is known whether the samples are repeatable and homogeneous at the beginning of the test.Only table 2 and 3 give specific values.This makes it difficult to understand the advisability of undertaking research. The authors presented a lot of research and analysis, but did not refer to the work of other authors when discussing their results, there is no discussion in this respect. This article is more about simulation techniques than polymer issues. The article may be published in the Polymers journal, but the authors need to make some changes to change the profile of the article.

Author Response

Dear respectful Reviewer, 

Many thanks for all your valuable remarks. All of them are reasonable and make it possible to improve the text substantially. Please find below the exact replies for all your notes. The corresponding corrections were put in the text of the manuscript and marked with cyan.

Thank you once again for all the efforts on improving the manuscript,

Sergey Panin

The authors should check the correctness of the chemical nomenclature as agreed by UPAC.

It was corrected! We have inserted IUPAC nomenclature for PTFE and PEEK, while it coincides with the used abbreviation for PPS. Also, commercial brand-names are given for PEEK and PPS resins that make it easier to identify their properties.

The entire introduction concerns methods and problems related to simulation and there is nothing about what these simulations have helped to achieve in the field of modern polymer composites.

Thanks a lot for your remark.

The idea of the manuscript has been formulated to a greater extent as the development of a methodology to design the polymer composites having both high strength and wear resistance. These two properties have been often achieved in opposite ways. At that, the authors have not used any “exclusive” fillers, primarily to illustrate the possibility of the methodology implementation. However, this does not exclude options to design some composites based on advanced polymer matrixes, compatibilizers, reinforcing inclusions or (nano)modifiers. In addition to mechanical and tribological properties, the composites can be designed in accordance with the requirements of biocompatibility, electrical or thermal conductivity, etc.

Table 1 gives the ranges of properties of composites, but it should be specified which indicators for materials with different fillings were concisely obtained, taking into account standard deviations so that it is known whether the samples are repeatable and homogeneous at the beginning of the test.

Thanks a lot for your remark.

The specified operational properties for the composites shown in Table 1 imply the achievement of all of them at the same time. This is the idea of the methodology to overlap all surfaces of the operational properties in a single graph. The paper discusses the aspects of improving the wear resistance of the PEEK-based composites, which are initially not wear-resistant. If these properties cannot be achieved within the used composition, then other requirements are reduced, or a new composition is designed. The reproducibility of the results has been confirmed by statistical data processing and testing of at least four samples of each composition. The uniformity of the structure has been controlled by analysis of the SEM micrographs, including for different sections of the sintered samples.

Only Tables 2 and 3 give specific values. This makes it difficult to understand the advisability of undertaking research.

Thanks a lot for your deep remark.

The idea of the proposed methodology for designing the three-component polymer composites is based not on the identification of strictly regulated contents of each filler, but on the search for ranges of their quantity. For this reason, the data in Tables 2 and 3 show that the specified operational properties will be achieved at the center of this range. In this investigation, it has successfully coincided that the pre-tested samples (compositions) corresponded to the center of the range. As a rule, a completely different composition is designed based on pre-calculated data and used to make samples then.

The authors presented a lot of research and analysis but did not refer to the work of other authors when discussing their results, there is no discussion in this respect.

Thanks a lot for your deep remark. We have added a couple of paragraphs and a dozen references in order to illustrate the place of the current study in regard to the recent publications on the topic.

The polymer composites having excellent tribological properties are widely used in various transport systems due to their high self-lubricating ability and temperature stability. In order to achieve the required operational properties, various types of fillers are loaded into the polymer composites in accordance with the functional tasks to be solved. Thus, reinforcing fibers (carbon, glass, aramid, etc.) [1–3], as well as solid lubricant fillers (primarily PTFE [6–8]) are usually used to simultaneously improve the mechanical and tribological properties of the PEEK- and PPS-based composites. The fibers improve the strength of the composites by increasing the elastic modulus, and solid lubricant particles reduce the friction coefficient and wear rate. V. Rodriguez et al. [9] have shown that loading of PEEK with graphite and PTFE in a content of 10 wt. % enabled to simultaneously improve strength (elastic modulus increases by 4.6 times) and tribological properties (the friction coefficient decreases from 0.40 to 0.28). Zang et al. [12] have tested the PPS-based composites loaded with SCF and graphite when friction has been under diesel lubrication conditions. Loading with the fillers has significantly reduced the friction coefficient and wear rate. Thus, published data usually describes the results of an increase in the elastic modulus (from tens of percent to units of times), as well as a decrease in the friction coefficient and wear rate (also up to several times).

In this paper, two issues have been solved. Firstly, the optimal contents of two fillers have been determined (each of the fillers ensured the achievement of the opposite functional requirements). Secondly, the wear patterns of the three-component polymer composites under conditions of metal-polymer and ceramic-polymer tribological contacts have been analyzed. In this case, the reasons for the change in wear resistance of both the polymer composites and the harder counterparts have been shown.

This article is more about simulation techniques than polymer issues. The article may be published in the Polymers journal, but the authors need to make some changes to change the profile of the article.

Thanks a lot for your remark.

You are absolutely right that the authors have focused more on description and verification of the model representations than on the design of the fundamentally new polymer composites. Currently, we have prepared a manuscript on the design of the “PEEK–nanoparticles–PTFE” three-component composites. It focuses on the patterns of structure formation and aspects of increasing wear resistance. According to the authors, computer simulation methods are also of interest to the polymer community. As part of the responses to the comments of the distinguished reviewer, a number of explanations have been added to take into account this remark.